# Hypergraph-based Temporal Modelling of Repeated Intent for Sequential Recommendation

## ABSTRACT

In sequential recommendation scenarios, user intent is a key driver of consumption behavior. However, consumption intents are usually latent and hence, difficult to leverage for recommender systems. Additionally, intents can be of repeated nature (e. g., yearly shopping for christmas gifts or buying a new phone), which has not been exploited by previous approaches. To navigate these impediments we propose the *HyperHawkes* model which models user sessions via hypergraphs and extracts user intents via contrastive clustering. We use Hawkes Processes to model the temporal dynamics of intents, namely repeated consumption patterns and long-term interests of users. For short-term interest adaption, which is more fine-grained than intent-level modeling, we use a multi-level attention mixture network and fuse long-term and short-term signals. We use the generalized expectation-maximization (EM) framework for training the model by alternating between intent representation learning and optimizing parameters of the long- and short-term modules. Extensive experiments on four real-world datasets from different domains show that HyperHawkes significantly outperforms existing state-of-the-art methods.

## CCS CONCEPTS

• **Information systems → Recommender systems**.

## KEYWORDS

Recommender Systems, Sequential Recommendation, Graph Neural Network, Hypergraph, Intent

**ACM Reference Format:**
Anonymous Author(s). 2024. Hypergraph-based Temporal Modelling of Repeated Intent for Sequential Recommendation. In *Proceedings of 2025 ACM Web Conference (WWW '25)*. ACM, New York, NY, USA, 11 pages. https://doi.org/10.1145/1122445.1122456

## 1 INTRODUCTION

Recommender systems have long become essential in filtering information effectively, for instance on video-sharing websites, e-commerce platforms, online bookstores, and social networks. With the abundance of online information, recommender systems have gained increasing importance by discovering and leveraging the underlying (latent) intents of users to cater to their preferences. In

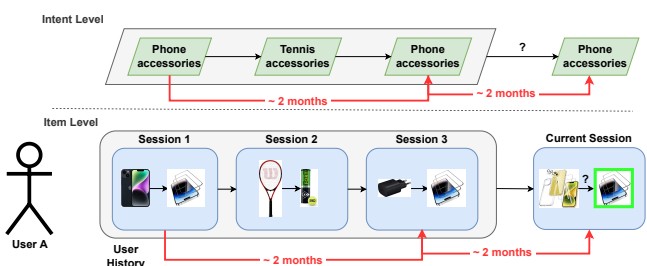

**Figure 1: A toy example of an e-commerce retailer scenario with repeated user intents.**

recent years, there has been a growing trend in modeling user sequential behaviors, which aims to capture short-term user interest and longer-term sequential patterns including popularity trends and interest drifts [42]. While traditional recommendation methods focus on static user preference modeling [16, 45], Sequential Recommendation (SR) models dynamically characterize user behaviors [18, 24], aiming to accurately predict users' interests in items based on their historical interactions and their corresponding points in time, allowing for more accurate and timely recommendations [8, 54].

The majority of previous works in SR order items by interaction timestamps and focus on sequential patterns to predict the next potential item. Early works adopt Markov chains to provide recommendations based on the $L$ previous interactions via an $L$-order Markov chain [15, 46]. Also, Recurrent Neural Networks (RNN) and Convolutional Neural Networks (CNN) have been applied to model long- and short-term dependencies in a user interaction sequence [18, 67]. More recent methods rely on the self-attention mechanism and transformer-based models for capturing complex sequential dependencies for next-item recommendations [24, 49]. Another line of work explicitly focuses on modeling temporal dynamics in item sequences based on interaction timestamps [31, 66]. The availability of temporal information also enables models to learn about global events (e. g., Christmas) [56] and the periodicity of items [4, 54]. Previous works in the field model the temporal dynamics on an item level or rely on additional category and knowledge-graph information to represent user intent [19, 55]. However, these approaches come with several downsides: Learning temporal dynamics on the item level is often difficult due to data sparsity and ignores co-occurring item consumption patterns across all users. Also, valuable meta-information for learning user intents is not always available and mostly ignores personal user preferences like preferred brands, price restrictions, or re-consumption behavior.

To fill the aforementioned gaps, we propose to extract latent user intents from the user interaction sequences and model personalized temporal dynamics including repeat consumption on the user intent level. Consider the example in Figure 1. During each session, User A interacts with the system by e.g., viewing or purchasing items

with different intents, and in this example, their interest is solely focused on the items relevant to their current intent. From the user's interaction history, it is apparent that the intent of consuming phone accessories is of repeated nature and is connected to the lifetime of a screen protector for the phone. Explicitly modeling this behavior increases the ability to recommend suitable phone accessories after a certain period (e. g., two months).

Repeat consumption occurs due to people's habits. For instance, we frequently purchase the same items, dine at the same restaurants, and listen to the same songs and artists often with a certain intent [1]. To empirically analyze the intent repeat consumption in the real world, we extract sets of frequently co-occurring item sets via the FP-Max algorithm [13]. For each active user (a user with at least 20 item interactions) we compute maximum frequent item sets (appearing twice or more in the user history) with a size larger than 1 to capture intent-level interactions. Then, we compute the maximum support of all repeated intents per user, where a support of 0.5 of an item set means this intent is apparent in 50% of the user's sessions. Figure 2 displays the distribution of intent repeat consumption with different maximum support values for four real-world benchmark datasets from different domains. Although there is a large portion of users with non-repeating intents, it is clear to see that intent repeat consumption is prevalent, and also constitutes a significant portion of interactions in certain domains.

To bridge this described gap of modeling temporal dynamics of user intents we propose the **Hyper**graph-based **Hawkes** Processes (**HyperHawkes**) model for sequential recommendation. Our approach leverages hypergraphs and soft clustering to extract latent user intent representations from the user interaction data. Based on these user intent representations our temporal excitation module learns the dynamics of user intents and item consumption behavior based on Hawkes Processes [14], a temporal point process to model discrete events in a continuous-time regime. We propose a novel time decay function to represent the excitation strength between historical intent and item behaviors and their corresponding time intervals. To capture short-term interest changes on the item level, we additionally compute short-term interest scores based on an attention mixture network, which captures the influence of the last interacted items in the current session. These steps ensure that our model effectively combines long-term and short-term user interest, and models both intent- and item-level temporal dynamics. We summarize our main technical contributions as follows:

- We propose a novel global item hypergraph construction strategy for learning intent-based item representations and employ soft clustering to extract latent user intents.
- We integrate Hawkes Processes (temporal point processes) to model long-term temporal dynamics on intent level; further, we fuse short-term interests for increased personalized recommendation performance.
- We conduct extensive experiments showing that our proposed model achieves significant performance improvements over a large number of state-of-the-art competitors on four datasets from different domains.[1]

---

[1]Code: https://anonymous.4open.science/r/HyperHawkes-2FB8

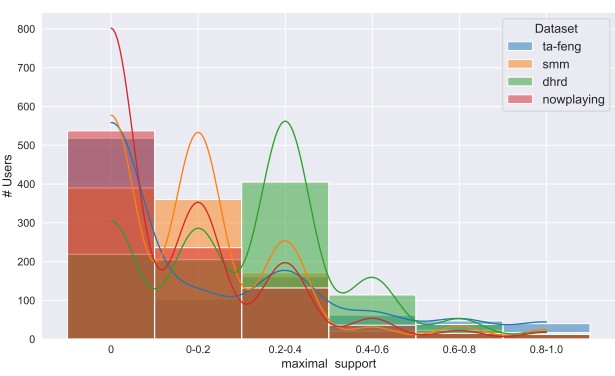

**Figure 2: Distribution of maximal support of intents (item sets with size >= 2) of active users per dataset. We randomly sampled 1000 users per dataset to ensure comparability between datasets.**

## 2 RELATED WORK

In this section, we review related work, which includes sequential recommendation, user intent, and temporal information learning.

### 2.1 Sequential Recommendation

Sequential recommendation aims to recommend items to the user by modeling their past behavior sequences and characterize their dynamic interests [24, 39, 42, 46]. Earlier approaches in this field are based on nearest-neighbor methods [12, 21], factorization machine-based methods [44] and Markov Chains [15]. In recent years the advances of deep learning also led to the deployment of many deep sequential recommendation models including CNN-based models [50, 67], RNN-based models [18, 65] and self-attention based models [10, 24, 49]. SASRec [24] and BERT4Rec [49] both utilize the transformer architecture [53] to model correlations among context information in SR. Recently, many works focused on using contrastive self-supervised learning (SSL) to enhance the mutual information between positive samples while increasing the discrimination of negatives [38, 41, 51, 63, 73].

### 2.2 User Intent for Recommendation

In recent times an increasing body of work studied users' intents for improving sequential recommendations [29, 30]. Works in session-based recommendation learn different purchase purposes via a mixture-channel purpose routing network [57], use a multi-intent translation graph neural network to mine user intents [35] or employ a dual-intent network to recommend new items [22]. Work in [71] proposes an attention mixture network based on user intents to achieve multi-level reasoning over item transitions. Another area of research focuses on understanding the sequential patterns in users' interaction behaviors over longer periods. DSSRec [36] introduces a seq2seq training strategy that utilizes multiple future interactions as supervision and incorporates an intent variable derived from both the user's past and future behavior sequences. In ICLRec [7] user intents are represented by latent variables and learned via clustering. The learned intents are leveraged into SR models via contrastive SSL to

maximize the agreement between the representation of a sequence and its corresponding intent.

## 2.3 Temporal Information, Repeated Consumption

Time-sensitive recommendation considers the temporal information of item interactions as context features or models temporal decay effects of historical interactions via point processes. In TimeSVD++ [62], timestamps are divided into bins and combined with a collaborative-filtering framework. In tensor factorization methods time is viewed as an extra dimension in the user-item interaction matrix [5, 25, 64]. Other works focus on capturing trends and user-evolving patterns via attention-based temporal modules [8, 9, 43, 66]. Li et al. [31] extend SASRec by modeling the user-specific time intervals in the item sequence. Recently, TGSRec [11] designs a continuous-time bipartite graph, which captures temporal dynamics within the sequential patterns of user-item interactions. Another line of work applies the Hawkes Process framework [14] to model the temporal decay effects of historical interactions [6, 74], which also increases the capability of the model to predict repeating item interactions [4, 19, 54, 55].

Different from previous works, we not only leverage that repeated interactions occur at intent levels but also show that incorporating personal user information is crucial for learning temporal dynamics. Additionally, our model addresses the gaps in the current understanding of user intents, especially in terms of capturing repeated and periodic patterns, modeling user intents through a hypergraph and soft clustering techniques based on user session information, which significantly enhances personalized recommendation performance.

## 3 PRELIMINARIES

### 3.1 Problem Definition and Notations

In sequential interaction scenarios, the observed user-item interaction data is represented by a set of tuples $\{(u, v, t)\}$, indicating that user $u \in \mathcal{U}$ interacted with item $v \in \mathcal{V}$ at timestamp $t$. The interactions are sorted chronologically to form a user's interaction sequence $I_u^t = [(v_1, t_1), (v_2, t_2) \dots, (v_n, t_n)]$, where $n$ is the number of interactions of user $u$ until timestamp $t$. Based on the varying time intervals between interactions, the sequence $S_u$ can be divided into subsequences (or sessions) whenever the time interval between two interactions exceeds a threshold $\delta$ (e. g., a day or hour). The resulting session interaction sequence can be represented as $S_u = [s_1^u, s_2^u, \dots, s_l^u]$, where $s_l^u$ represents the $l$-th interaction subsequence of user $u$ containing items from $\mathcal{V}$. The objective of sequential recommendation is to predict the item from the item set $\mathcal{V}$ that the user $u$ is most likely to interact with at a given timestamp $t$, given their sequence $S_u$.

### 3.2 Hawkes Processes for Sequential Modeling

A temporal point process is a stochastic process consisting of discrete events localized in the continuous-time domain. In sequential recommendation, the times at which a user interacts with a specific item can be represented as a series of historical events $H_t = [t_1, t_2, \dots, t_n]$. To model the time of the next event based on previous events, a conditional intensity function $\lambda(t|H_t)$ is introduced. This function represents a stochastic model for the occurrence of the next event given all previous event times and thereby, affects the characteristics of the temporal point process. In Hawkes Processes [14], the intensity function takes the form of

$$\lambda(t) = \lambda_{Base} + \alpha \sum_{t_i < t} \varphi(t - t_i), \tag{1}$$

where $\lambda_{Base}$ represents the base intensity and each historical event has a self-exciting effect on the current intensity controlled by the triggering kernel $\varphi$ which determines how each past event boosts the event intensity over time. The parameter $\alpha$ determines the degree of excitation. In the context of sequential recommendation, the base intensity represents the user's basic interest in a target item, and the self-exciting term indicates the cumulative impact of historical interactions on the user's interest over time.

## 4 PROPOSED METHOD (HYPERHAWKES)

As illustrated in Figure 3 our HyperHawkes model consists of several major components, including the intent-based global item graph, and a hypergraph-based aggregation layer to generate intent-based item representations for the soft clustering component. The clustered intent-based item representations serve as inputs to the temporal module, which is responsible for capturing users' long-term interests. To model short-term interest we employ an attention-mixture network and combine both long-term and short-term signals in the final prediction layer. In the following, we will detail each component.

### 4.1 Intent-based Hypergraph Network

As user intents are latent by definition and hence are difficult to extract, we propose to induce structural bias via hypergraph modeling to support the underlying soft clustering process to find useful intent representations. Compared to a simple graph with an adjacency matrix reflecting the pairwise relationship between two nodes, hyperedges in hypergaphs can connect more than two nodes and are therefore suitable to model user intents, since item interactions on an intent level naturally comprise a set of items. We assume that in each user session, the user interacts with the system based on one or more intents. To build our intent-based global item hypergraph $\mathcal{G} = (\mathcal{V}, \mathcal{E})$ with $\mathcal{E} = \{\varepsilon_i\}$ being the set of hyperedges, we apply the following procedure: First, we extract data-driven user intents as frequently occurring item sets across all training user sessions with a length $>= 2$ via the FP-Max algorithm [13], where the minimum support is set to $\gamma$. The threshold parameter $\gamma$ filters for reliable user intents and drops noisy intents not supported by many other user sessions [38]. For each of the extracted intents, we connect all the corresponding items via a hyperedge $\varepsilon_i \in \mathcal{E}$ to build our global hypergraph. Each hyperedge $\varepsilon_i$ has a weight $w_i$ attached, indicating the frequency of the extracted intent in the dataset.

To generate intent-based global item representations we design a simple hypergraph aggregation layer. For the item $v_i$'s base embedding $\mathbf{x}_i^{(0)}$, we map its corresponding identifier into a dense embedding vector $\mathbf{h}_{v_i} \in \mathbb{R}^d$, where $d$ indicates the dimension. To aggregate information from neighboring nodes we employ the following hypergraph convolution with symmetric normalization in our HGCN component:

$$\mathbf{X}^{(l+1)} = \mathbf{D}^{-1}\mathbf{H}\mathbf{W}\mathbf{B}^{-1}\mathbf{H}^{\top}\mathbf{X}^{(l)}, \tag{2}$$

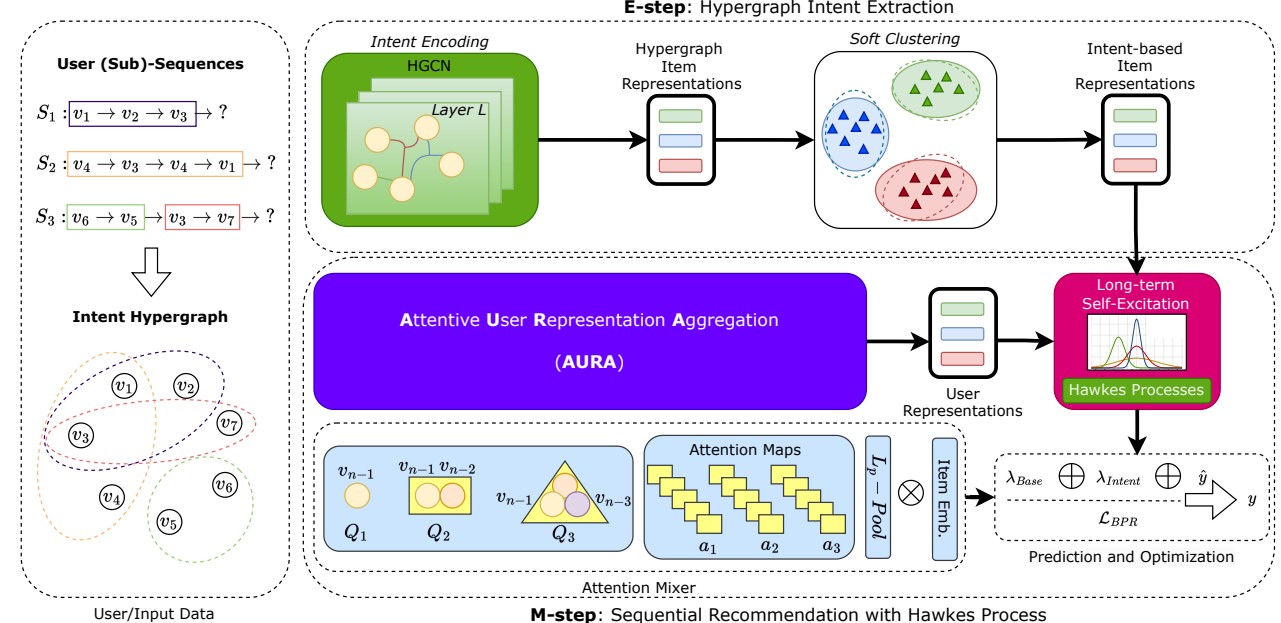

**Figure 3: Overall architecture of HyperHawkes: In the E-step of the EM algorithm, our approach extracts latent intent representations via soft clustering of hypergraph-based item embeddings. In the M-step, we compute long-term user preference scores via Hawkes Processes based on the user base excitation from an attentive FISM and self-exciting effects of intents. We fuse short-term scores from the attention-mixture network and the long-term scores to get the preference score of the user for an item.**

where $\mathbf{H}$ is the incidence matrix, $\mathbf{W}$ is the diagonal hyperedge weight matrix, and $\mathbf{D}$ and $\mathbf{B}$ are the corresponding degree matrices. Compared to the hypergraph convolution presented in [3] we do not make use of learnable weights and a non-linear activation function, since these components are not essential for recommender systems [59, 61]. To combine node embeddings over multiple layers and increase the receptive field of a node we average the node embeddings over $L$ layers to get the final intent-based hypergraph item representations:

$$\mathbf{X}^{(L)} = \frac{1}{L+1} \sum_{l=0}^{L} \mathbf{X}^l. \tag{3}$$

## 4.2 Intent Representation Learning

On the user interaction sequence level, it is easily observed that user sessions exhibit multiple, dynamically shifting intents, where items can also belong to more than one specific intent alone [7, 48]. Additionally, these intents are not confined solely to individual sessions but are also prevalent among users with similar preferences. Therefore, directly utilizing session representation distributions for intent representations will result in a loss of information. To mitigate this, we introduce a soft clustering component to disentangle latent intents and effectively cluster items to intents.

For our soft clustering component we adopt a soft version of the Lloyd's $k$-means algorithm [58]. Let $\mathbf{x}_j$ represent the intent-based hypergraph representation $\mathbf{x}_j^{(L)}$ of item $v_j$ and $\mu_k$ represent the center of intent cluster $k$. The variable $r_{jk}$ denotes the probability

to which item $v_j$ is assigned to intent cluster $k$. In the standard $k$-means algorithm, this assignment is binary, but we relax it to allow fractional values such that $\sum_k r_{jk} = 1$ for all $j$. Specifically, we define

$$r_{jk} = \frac{\exp(-\beta \|\mathbf{x}_j - \mu_k\|)}{\sum_\ell \exp(-\beta \|\mathbf{x}_j - \mu_\ell\|)}, \tag{4}$$

which provides a soft-min assignment of each point to the cluster centers based on distance. We use negative cosine similarity as a distance norm $\|\cdot\|$. Here, $\beta$ is an inverse-temperature hyperparameter; taking $\beta \to \infty$ recovers the standard $k$-means assignment. The intent cluster centers can be optimized via an iterative process similar to the traditional $k$-means updates by alternately setting

$$\mu_k = \frac{\sum_j r_{jk} \mathbf{x}_j}{\sum_j r_{jk}} \quad \forall k = 1, \dots, K \tag{5}$$

and

$$r_{jk} = \frac{\exp(-\beta \|\mathbf{x}_j - \mu_k\|)}{\sum_\ell \exp(-\beta \|\mathbf{x}_j - \mu_\ell\|)} \quad \forall k = 1, \dots, K, \; j = 1, \dots, n. \tag{6}$$

These iterations converge to a fixed point where $\mu$ remains unchanged between successive updates. As a result, we have soft intent cluster assignments for each item $\mathbf{p}_j \in \mathbf{P}$ corresponding to probabilities that item $v_j$ belongs to one of the intent clusters $K$. This probability distribution $\mathbf{p}_j$ serves as the latent intent representation of item $v_j$.

Since the intent representations $\mathbf{p}_j \in \mathbf{P}$ are latent by definition we face the issue that without the cluster representations, we cannot estimate our model parameters $\theta$ and without $\theta$ we are not able

to find a result for the soft cluster assignment probabilities $\mathbf{P}$. It has been shown that a generalized Expectation-Maximization (EM) framework can resolve this situation [7, 34]. In its basic idea, EM starts with an initial guess of $\theta$ and estimates the expected values of our cluster assignments $\mathbf{P}$ in the E-step. In the M-step we maximize the objective w.r.t. the model parameters $\theta$ given the expected values of $\mathbf{P}$. These steps are repeated until the likelihood cannot increase anymore. For detailed derivations of the EM framework under the sequential recommendation scenario we refer to [7, 34].

### 4.3 Repeated Long-term Intent Consumption

We employ Hawkes Processes to model the temporal dynamics of long-term interactions on intent level. As defined in Equation 1 $\lambda_{Base}$ reflects the long-term base interest of a user in a specific item at a given point in time $t$, whereas the second part accounts for the self-exciting effects $\lambda_e$ and can capture repeated intent behaviors. We detail these two components in the following.

*4.3.1 User Base Preference.* Users often have diverse or even contrastive preferences (e. g., romantic and horror movies). Hence, using a single embedding vector to represent the long-term user interest is a limiting factor [60]. Previous works mitigate this issue by generating a global and non-causal representation of each user interaction sequence. Previous works [23, 33] built the preference representation of a user for an interacted item by a uniform aggregation of the representation of the other items in the interaction sequence. In our approach, we incorporate an attentive user representation aggregation (AURA) to compute the basic strength of the Hawkes Process $\mu$ which computes user representations flexibly based on the current target item representation $\mathbf{h}_v$:

$$\lambda_{Base}(u,v) = \mathbf{h}_u + \sum_{j \in I_u \setminus \{v\}} \frac{\exp(\mathbf{h}_j^\top \mathbf{h}_v)}{\sum_{j' \in I_u \setminus \{v\}} \exp(\mathbf{h}_{j'}^\top \mathbf{h}_v)} \mathbf{h}_v \quad (7)$$

where $\mathbf{h}_u \in \mathbb{R}^d$ defines the latent user representation and is fused with the long-term preference of user $u$ for item $v$ which is a weighted aggregation of the item representations in the user interaction sequence $I_u$.

*4.3.2 Intent Excitation Learning.* The trigger kernel of the intensity function in the Hawkes Process captures the changing excitation over time. Our goal is to leverage the time dynamics of a user's next intent and how previous intents can trigger subsequent interactions. The Hawkes Process simulates the time dynamics to predict the probability of the next event. In our approach, we consider interaction events with the same underlying intent for self-excitation. Particularly, we define intent excitation learning as follows:

$$\lambda_{Intent}(u,v,t) = \alpha_k \sum_{(v',t') \in I_u^t} I_K(\mathbf{p}_v, \mathbf{p}_{v'}) \varphi(t-t') \quad (8)$$

where $I_K$ denotes the indicator function which returns 1 if item $v$ and $v'$ belong to the same intent cluster and are in different user sessions, otherwise it returns 0. Since we use a soft clustering approach to assign intent cluster probabilities to each item we use the Kullback–Leibler divergence for finding items that correspond to the same intent clusters:

$$I_K(\mathbf{p}_v, \mathbf{p}_{v'}) = \sum_{x \in \mathcal{X}} \mathbf{p}_v \, \log\left(\frac{\mathbf{p}_v}{\mathbf{p}_{v'}}\right) > \delta, \quad (9)$$

where $\delta$ is a parameter to limit the probability distribution distances per intent cluster assignment. The cluster-related parameter $\alpha_k$ weights the degree of excitation. The temporal kernel function $\varphi(\cdot)$ changes with the time interval $\Delta t = t - t'$ between items of the same intent and is defined as:

$$\varphi(\Delta t) = (1 - \pi_k)E(\Delta t | 1/\beta_k) + \pi_k N(\Delta t | \mu_k, \sigma_k), \quad (10)$$

where we leverage an exponential distribution with intent-based parameter $\beta_k$ to model short-term intent repeat consumption behavior, which diminishes quickly over time. For long-term repeated behavior we employ a normal distribution with mean $\mu_k$ and standard deviation $\sigma_k$ which are also intent representation-based parameters. Using normal distributions to simulate the user dynamic interest changes captures real-world scenarios like item lifecycles and repeated item consumption behavior [19, 54]. The coefficient $\pi_k$ balances the two distributions. We learn the corresponding parameters of the distributions $\Theta_{Intent} = \{\alpha_k, \beta_k, \mu_k, \sigma_k, \pi_k\}$ by a non-linear transformation of the user representation $\mathbf{h}_u$, item representation $\mathbf{h}_v$ and the intent representation $\mathbf{p}_v$:

$$\Theta_{Intent} = \mathcal{M}(\mathbf{h}_u || \mathbf{h}_v || \mathbf{p}_v), \quad (11)$$

where $\mathcal{M}(\cdot)$ is implemented as a two-layer neural network and $||$ denotes the vector concatenation operation. Compared to previous approaches [19, 54] our distribution parameters are not related to item identifiers, but to the corresponding item and intent representations. Hence, our model learns the temporal dynamics on both, item and intent level, and is able to leverage denser input signals, since the number of intents is usually smaller than the number of items in a dataset. Additionally, the incorporation of the user representation to compute the distribution parameters allows our model to learn user-specific repetition behavior which can vary across intents. For instance, one user buys a new phone including accessories every year whereas another user only buys a new phone if the old one is broken and therefore has a longer intent cycle phase.

We introduced the base intensity $\lambda_{Base}(u,v)$ as well as the long-term self-excitations $\lambda_{Intent}(u,v,t)$ on intent level. Therefore, we define our final long-term excitation for item $v_i$:

$$\lambda_i(u,v_i,t) = \lambda_{Base}(u,v_i) + \lambda_{Intent}(u,v_i,t) \quad (12)$$

### 4.4 Attention Mixtures for Short-term User Interest

The aforementioned components model the long-term interest and repeated consumption behavior of a user based on intents. However, a user intent might be of an exploratory nature, or the interest may change dynamically during the session. To capture these dynamics of short-term user interest, we employ an attention mixture mechanism, which has shown to be a promising approach for session-based as well as sequential recommendation [52, 71]. Following [71] we generate multi-level intent queries on the groups of last items in a user interaction sequence with length $n$ by employing the deep sets operation [68] and applying linear transformations per level $m \in M$:

$$\mathbf{Q}_M = \mathbf{W}_M\left(\sum \{\mathbf{h}_{v_i}\}_{i=n,\dots,n-M+1}\right). \quad (13)$$

These generated queries are then used to compute multi-head attention weights as:

$$\alpha_h = \text{softmax}\left(\frac{\mathbf{Q}\mathbf{W}_Q(\mathbf{K}\mathbf{W}_K)^\top}{\sqrt{d}}\right), \qquad (14)$$

where $\mathbf{Q} \in \mathbb{R}^{l \times d}$ is the query matrix, $\mathbf{K} \in \mathbb{R}^{n \times d}$ represents the hidden representation of each item in the sequence and $\mathbf{W}^Q, \mathbf{W}^K \in \mathbb{R}^{d \times d}$ are trainable parameters. We apply $L_p$-pooling [20] to pool the attention map and multiply the hidden representation of the items in the sequence with the corresponding pooled attention weights to get the final short-term sequence representation $\mathbf{s}_u$.

## 4.5 Prediction and Model Optimization

For the next-item prediction task we need to combine long-term and short-term interests of users. We use the short-term sequence representation $\mathbf{s}_u$ to compute the short-term interest score $\hat{y}_i = \mathbf{s}_u^\top \mathbf{h}_{v_i}$, for item $v_i$. Then, we add the long-term excitation score $\lambda_i$ and the short-term interest score $\hat{y}_i$ to get the final preference score:

$$y_i = \lambda_i + \hat{y}_i. \qquad (15)$$

To learn the parameters of our recommendation model in the M-step of the EM algorithm we adopt the pairwise ranking (BPR) loss for optimization as follows:

$$\mathcal{L}_{BPR} = -\sum_{u \in \mathcal{U}} \sum_{i=1}^{N_u} \log \sigma(y_{ui} - y_{uj}), \qquad (16)$$

where $\sigma$ denotes the sigmoid function and $y_{uj}$ reflects the preference score of user $u$ to a randomly sampled negative item $j \notin I_u^t$.

## 5 EXPERIMENTS AND RESULTS

In this section, we provide the setup and results of extensive experiments to evaluate our proposed model, where we compare Hyper-Hawkes to various state-of-the-art models in SR. Given our overall goal of investigating the impact of intent repeat-consumption and fusing short- and long-term interests of users, we aim to answer the following research questions:

- **RQ1:** How does our proposed HyperHawkes compare to other state-of-the-art SR methods on different datasets?
- **RQ2:** How do different components in HyperHawkes contribute to the performance?
- **RQ3:** How sensitive is HyperHawkes to different hyperparameter settings (e. g., $L$, $K$)?

## 5.1 Experimental Setup

*5.1.1 Datasets and Preprocessing.* To evaluate the performance of our approach, we conduct experiments on four representative datasets from the e-commerce, food delivery, and music domains [17, 27]. *Ta-Feng*[2] is a dataset containing Chinese grocery store transaction data from 2001. *SMM*[3] chronicles user behavior captured over the span of five months from January 15 to May 15, 2023 from a large online store [47]. For this industrial-scale dataset, we sample 20,000 random users to maintain consistency with the other datasets. The *DHRD* (Delivery Hero Recommendation Dataset)[4] is

a dataset presented in [2] and comprises food delivery orders from three distinct cities, encompassing different vendors and dishes. For our evaluation, we use the data related to the city of Stockholm. Lastly, the *NowPlaying* dataset comes from the music domain and is described in [70]. This dataset includes music listening behavior of users based on Twitter data. It is worth noting, that we do not provide evaluation for the widely used Amazon review datasets [37], the MovieLens datasets[5], or the Yelp review datasets[6], since those datasets are rating/review-based and therefore do not include repeated item consumptions, making them unsuitable for the scenario of repeated intent modeling [17, 27].

**Table 1: Dataset statistics (after preprocessing): Number of users, items, interactions, average sequence length and sparsity.**

| Dataset | Ta-Feng | SMM | DHRD | NowPlaying |
|---|---|---|---|---|
| $|\mathcal{U}|$ | 26,162 | 12,098 | 42,774 | 11,310 |
| $|\mathcal{V}|$ | 15,642 | 22,167 | 20,883 | 15,905 |
| # Interactions | 0.78m | 0.87m | 0.52m | 1.12m |
| Avg. length | 29.99 | 71.97 | 12.30 | 86.39 |
| Sparsity | 99.80% | 99.67% | 99.94% | 99.45% |

We follow the preprocessing steps as shown in [7, 19] for the four datasets. To be more specific, we keep the *5-core* datasets, where users and items with less than 5 interactions are filtered out. Table 1 provides an overview of the datasets after preprocessing. To split the datasets, we follow common practice in sequential recommendation and use interactions with the second latest time for validation and interactions with the latest timestamp for testing.

*5.1.2 Evaluation Metrics.* Following previous works [17, 28, 69], we use the whole item set without negative sampling to rank the predictions. We adopt *HR@{5,20}* (Hit Ratio) and *NDCG@{5,20}* (Normalized Discounted Cumulative Gain) to evaluate the quality of the recommendation results.

*5.1.3 Baseline Methods.* We compare HyperHawkes with the following representative baseline and state-of-the-art methods for sequential recommendation:

**Static models**: BPR-MF [45] is a non-sequential model and characterizes the pairwise interactions via matrix factorization.

**Standard sequential & Transformer models**: As standard sequential models we include GRU4Rec [18], an RNN-based method and SASRec [24] as transformer-based baseline method for SR.

**Temporal & intent-based models**: SLRC [54] is a widely used model and one of the first to model item repeat consumption. It combines matrix factorization with a temporal point process, effectively capturing short-term and product lifetime effects. RepeatNet [43] proposes a novel repeat-explore mechanism to balance repeated and new item consumption. For intent-based methods, we include HIDE [32] which models intents via session hypergraphs. Other state-of-the-art approaches include ICLRec [7] and ICSRec [40], where user intents are learned via clustering and Atten-Mixer [71], where intents are modelled via a multi-level network.

---

[2]https://www.kaggle.com/retailrocket/ecommerce-dataset
[3]https://disk.yandex.ru/d/fSEBIQYZusAAuw/datasets/data_smm
[4]https://github.com/deliveryhero/dh-reco-dataset

[5]https://grouplens.org/datasets/movielens
[6]https://www.yelp.com/dataset

**Table 2: Model performance on all four datasets (± standard deviation for HyperHawkes). All improvements of HyperHawkes over the second best model are significant (paired *t*-test, $p < .05$), best results are in boldface and the second-best results are underlined.**

| Dataset | Metric | BPR-MF | GRU4Rec | SASRec | SLRC | RepeatNet | HIDE | ICLRec | Atten-Mixer | ICSRec | HyperHawkes | Improv. |
|---|---|---|---|---|---|---|---|---|---|---|---|---|
| Ta-Feng | HR@5 | 0.0699 | 0.0657 | 0.0812 | 0.0714 | 0.0432 | 0.0616 | 0.0746 | 0.0878 | 0.0784 | **0.1108**±0.0015 | 26.19% |
| | HR@20 | 0.0943 | 0.1215 | 0.1629 | 0.1284 | 0.1006 | 0.0853 | 0.1415 | 0.1645 | 0.1566 | **0.1984**±0.0030 | 20.60% |
| | NDCG@5 | 0.0541 | 0.0459 | 0.0528 | 0.0488 | 0.0307 | 0.0419 | 0.0527 | 0.0605 | 0.0519 | **0.0765**±0.0014 | 26.44% |
| | NDCG@20 | 0.0610 | 0.0616 | 0.0761 | 0.0650 | 0.0469 | 0.0485 | 0.0716 | 0.0823 | 0.0742 | **0.1015**±0.0015 | 23.32% |
| SMM | HR@5 | 0.0542 | 0.0586 | 0.0876 | 0.1170 | 0.1291 | 0.0391 | 0.0526 | 0.0817 | 0.0686 | **0.1483**±0.0019 | 14.87% |
| | HR@20 | 0.1056 | 0.1323 | 0.1687 | 0.1853 | 0.1968 | 0.0781 | 0.1101 | 0.1638 | 0.1505 | **0.2444**±0.0009 | 24.19% |
| | NDCG@5 | 0.0373 | 0.0393 | 0.0606 | 0.0840 | 0.0972 | 0.0272 | 0.0357 | 0.0561 | 0.0427 | **0.1018**±0.0002 | 4.73% |
| | NDCG@20 | 0.0516 | 0.0602 | 0.0836 | 0.1037 | 0.1175 | 0.0383 | 0.0520 | 0.0793 | 0.0656 | **0.1294**±0.0004 | 10.13% |
| DHRD | HR@5 | 0.2156 | 0.1439 | 0.2065 | 0.2775 | 0.2702 | 0.1878 | 0.2554 | 0.2211 | 0.2129 | **0.2982**±0.0055 | 7.45% |
| | HR@20 | 0.3805 | 0.3214 | 0.4651 | 0.4158 | 0.3211 | 0.2625 | 0.4544 | 0.4295 | 0.4715 | **0.4830**±0.0019 | 2.43% |
| | NDCG@5 | 0.1488 | 0.0946 | 0.1303 | 0.2031 | 0.1983 | 0.1356 | 0.1740 | 0.1489 | 0.1323 | **0.2089**±0.0031 | 2.85% |
| | NDCG@20 | 0.1963 | 0.1450 | 0.2039 | 0.2430 | 0.2142 | 0.1572 | 0.2312 | 0.2084 | 0.2145 | **0.2621**±0.0069 | 7.86% |
| NowPlaying | HR@5 | 0.1272 | 0.0992 | 0.1229 | 0.1756 | 0.1765 | 0.1079 | 0.1654 | 0.1475 | 0.1375 | **0.1842**±0.0011 | 4.36% |
| | HR@20 | 0.2730 | 0.2327 | 0.2715 | 0.3117 | 0.2996 | 0.1984 | 0.3135 | 0.3011 | 0.2931 | **0.3526**±0.0006 | 12.47% |
| | NDCG@5 | 0.0879 | 0.0650 | 0.0802 | 0.1197 | 0.1217 | 0.0787 | 0.1156 | 0.1028 | 0.0929 | **0.1242**±0.0007 | 2.05% |
| | NDCG@20 | 0.1289 | 0.1025 | 0.1221 | 0.1589 | 0.1581 | 0.1042 | 0.1574 | 0.1462 | 0.1368 | **0.1713**±0.0008 | 8.43% |

*5.1.4 Implementation Details.* For a fair comparison, we rely on the RecBole framework [72] to implement our approach, using the provided implementations of the baseline models or re-implementing them accordingly. For all baseline models and our model, the embedding size is set to 64 and the batch size to 512. We do not limit the number of training epochs per model, but adopt an early-stopping strategy, which stops training after five consecutive rounds of performance decrease on the validation set. Each baseline model is optimized according to its corresponding hyperparameters.

For the optimization of HyperHawkes, we use Adam [26] with a learning rate of 0.001. The number of layers $L$ in the HGCN component and number of intent clusters $K$ are searched in the ranges of $\{1, 2, 3, 4, 5\}$ and $\{2, 4, \ldots, 128\}$ respectively. For the attention mixture network, we search the number of heads in the range of $\{1, 2, 4, 8\}$ and the number of levels $M$ in $\{1, 2, \ldots, 10\}$. The threshold parameters $\gamma$ and $\delta$ are set to 5e−4 and 1e−12, correspondingly. Our implementation is based on PyTorch 1.13.1 and Python 3.8.16. All experiments are performed on a workstation with an AMD Ryzen 2950X, a GeForce RTX 2070, and 256 GB main memory. We publish the code and the pre-processed datasets on GitHub[7].

## 5.2 Performance Comparison (RQ1)

In Table 2 we report the results of the the performance comparison of HyperHawkes and the proposed baselines. Surprisingly, the results do not affirm that sequential models generally outperform non-sequential methods since BPR-MF shows competing performance compared to GRU4Rec or SASRec. This displays the importance of learning temporal dynamics of repeated user behavior and the incorporation of user intent.

Advanced time-sensitive sequential models often incorporate additional temporal signals to augment recommendation performance. For instance, TiSASRec integrates both the item positions and time intervals in a sequence, yielding superior performance than its transformer-based counterpart SASRec. We further observe that

leveraging contrastive SSL in transformer-based architectures can improve performance, as exhibited by ICLRec which optimizes sequence representations via contrastive SSL at the user intent level. Also, the other intent-based method Atten-Mixer shows significant performance gains over standard sequential models. Among the baseline methods, SLRC and RepeatNet exhibit improved performance even over more sophisticated temporal and intent-based models, underpinning their robustness in recommendation tasks and their ability to model item repeat consumption.

Our model, HyperHawkes, triumphs over all other methods across all datasets, marking a significant advancement in the domain. The average improvements compared with the best baseline per dataset range from 2.43% to 24.19% in HR@20 and from 7.86% to 23.32% in NDCG@20. We attribute this increase in performance to the ability of our approach to effectively model long-term intent repeat behavior and short-term user interest, which we show in detail in our ablation study.

In terms of efficiency and model complexity, we report the training time per epoch on the *Ta-Feng* dataset as a practical proxy for model complexity. Intent-based models like HIDE, ICLRec, Atten-Mixer and ICSRec require 2231.63, 254.21, 13.10 and 174.17 seconds/epoch, respectively. SLRC and RepeatNet, which focus on repeat consumption, need 13.31s and 29.64s, correspondingly. Our HyperHawkes takes 27.75s per epoch on training and therefore, is more efficient than most of the other sequential neural network models, while substantially outperforming these models in recommendation performance. A similar trend in model complexity is also seen for the other datasets.

## 5.3 Ablation Study (RQ2)

HyperHawkes contains several components including a hypergraph-based graph convolutional network (HGCN), soft clustering (SC), user base interest (LT-UE), intent excitation learning (LT-SINE), and a short-term attention mixture network (ST-ATM). To verify the effectiveness of each component, we conduct an ablation study

---
[7]https://anonymous.4open.science/r/HyperHawkes-2FB8

**Table 3: Ablation study of HyperHawkes. The symbol ↓ indicates a performance drop of more than 10%, ND = NDCG.**

| Dataset Model | Ta-Feng | | NowPlaying | |
|---|---|---|---|---|
| | HR@20 | ND@20 | HR@20 | ND@20 |
| (A) w/o LT-SINE | 0.1632↓ | 0.0842↓ | 0.3331 | 0.1637 |
| (B) w/o LT-UE | 0.1818 | 0.0911↓ | 0.3241 | 0.1570 |
| (C) w/o HGCN | 0.1901 | 0.0969 | 0.3145↓ | 0.1544↓ |
| (D) w/o SC | 0.1732↓ | 0.0867↓ | 0.3377 | 0.1666 |
| (E) only ST-ATM | 0.1668↓ | 0.0841↓ | 0.2954↓ | 0.1451↓ |
| (F) w/o ST-ATM | 0.0914↓ | 0.0558↓ | 0.3314 | 0.1625 |
| HyperHawkes | **0.1984** | **0.1015** | **0.3526** | **0.1713** |

on two datasets. Table 3 displays the result of the ablation study on the *Ta-Feng* and *NowPlaying* datasets. These two datasets were chosen due to their different domains and characteristics in terms of repeat consumption (e. g., e-commerce vs. music streaming). From (A) and (B) we can see the impact of different components in the Hawkes Process for modeling temporal dynamics. Eliminating the intent excitation learning (A) or the user base preference (B), each notably diminishes the performance of the recommendation model to a similar extent. This shows the importance of extracting latent intents and modeling repeat behavior on the intent level compared to the item level only. We also investigate the effect of our proposed hypergraph-based network in (C), where removing the component also leads to a significant performance drop. This backs our assumption that inducing structural bias through the HGCN supports the soft clustering process and leads to more representative cluster/intent representations. Similar effects can be observed when dropping the soft clustering component in (D) and using a standard $k$-means instead, which showcases the benefit of disentangling user intents via soft probability distributions. Lastly, we explore the effects of the short-term attention mixture network. Relying only on the short-term component without any consideration of long-term effects (E) results in a noticeable performance drop. Dropping the short-term component (F) from HyperHawkes shows a substantial decline compared to the full model, reflecting the critical role of short-term user behavior understanding. The incorporation of both short-term and long-term effects leads to the best overall performance. The ablation study results for the other two datasets *SMM* and *DHRD* are consistent with these findings, but are not reported due to space constraints.

## 5.4 Impact of Hyper-Parameters (RQ3)

In this section, we investigate the impact of different hyper-parameters. We focus on the number of layers $L$ in the HGCN and the number of intent clusters $K$, since these hyper-parameters are related to the intent excitation learning, which has shown to have the highest impact on the performance of the final model (see Section 5.3). Figure 4a shows the performance of our model with different settings of layers $L$ on the *Ta-Feng* and *NowPlaying* datasets. A higher number of layers in the hypergraph-based network does not necessarily lead to an increase in performance due to oversmoothing, where node representations converge to the same values. We can find a sweet spot layer setting $L$ of 3 (*Ta-Feng*) and 2 (*NowPlaying*).

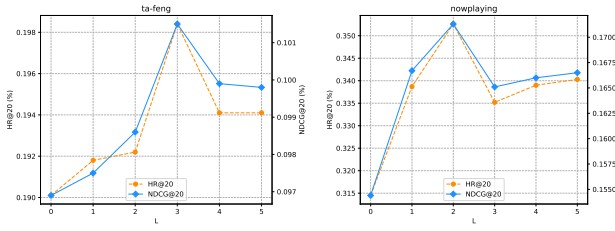

**(a) Different number of layers $L$ in our HGCN component.**

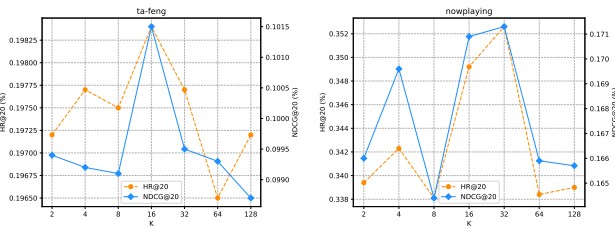

**(b) Number of intent clusters parameter $K$.**

**Figure 4: Impact of hyper-parameters in HyperHawkes.**

Our main contribution comprises the temporal modeling of user intents, where those user intents are extracted by soft clustering. Therefore, we have to define the number of clusters $K$ before the training of the model. Given the heterogeneity of datasets, this hyper-parameter needs to be carefully tuned to extract useful intent representations, or in other words, $K$ needs to be tailored to the characteristics of each dataset individually. Figure 4b shows the performance differences between runs with a different number of intent clusters. As the datasets stem from completely different domains, their best-performing setting also vary. For *Ta-Feng* its 16 clusters and for *NowPlaying* the best setting is 32 clusters.

## 6 CONCLUSION

In this paper, we propose HyperHawkes, a novel Hypergraph-based Hawkes Process model to comprehensively model temporal dynamics of user intents for generating personalized sequential recommendations. We extract intent representations via soft clustering of hypergraph-based item representations. Our model learns the long-term excitation of intents and items via Hawkes Processes and models short-term interests of users via a custom attention mixture component. The corresponding user preference scores from the long-term and short-term components are fused to provide temporal and personalized recommendations. The steps of finding clusters and learning temporal dynamics are alternately optimized under a generalized EM framework. Our extensive experimental results on four real-world datasets demonstrate the effectiveness of our proposed model over state-of-the-art methods. Our approach outperforms all other state-of-the-art methods, which only model repeated consumption on item level or use intents for contrastive learning purposes, in each of the provided metrics. The ablation study showed the impact of each component and that modeling repeat consumption is more important than focusing on short-term interest shifts of users. In future work, we aim to explore the temporal aspects of the extracted user intents for explainability purposes.

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
