# OpenReview forum: "Hypergraph-based Temporal Modelling of Repeated Intent for Sequential Recommendation"
_ACM.org/TheWebConf/2025/Conference — WWW 2025 Poster_

### Official Review · Reviewer_wEjM · 2024-11-06

**Novelty:** 5
**Technical Quality:** 5

**Review:**

This paper proposes the HyperHawkes model to model user sessions via hypergraphs and extract user intents via contrastive clustering.

The strengths are as follows. (1) The paper innovatively utilizes Hawkes Processes and EM algorithms to learn user intents. (2) The authors provide source codes for reproduction. Besides, the authors provide details for the experiment setting and hyperparameter study, which is good for reproduction.  (3) The experimental results on the proposed model and the *selected* baselines suggest that the proposed model is effective.

However, several issues exist in the paper. (1) The presentation of the paper needs improvement. For example, the variable ‘chi’ in Eq. (9) is not defined. The texts in Figure 4 are too small. (2) The authors need to include more recent *sequential* recommendation models for comparison. The Atten-Mixer model and HIDE are session-based recommendation models, which do not utilize user information. For a fairer and more comprehensive evaluation, the authors should include more recent sequential recommendation models as baselines.

**Questions:**

(1)	The authors need to clarify all ill-defined notations such as the variable ‘chi’ in Eq. (9). If these variables are indeed defined elsewhere, the authors should rearrange the paragraphs or provide a notation table for better clarity. (2) Since the EM algorithm potentially is less efficient, could the authors provide more efficiency experiments (e.g., number of epochs needed in the training phase, inference time, results on more datasets)?

**Reviewer Confidence:**

4: The reviewer is certain that the evaluation is correct and very familiar with the relevant literature

**Scope:**

4: The work is relevant to the Web and to the track, and is of broad interest to the community

---

### Official Review · Reviewer_i8UJ · 2024-12-01

**Novelty:** 4
**Technical Quality:** 4

**Review:**

Summary:
This paper presents the HyperHawkes model for sequential recommender systems, which aims to better capture repetitive user intent and its temporal dynamics. By using a hypergraph to represent user sessions and the Hawkes process to model the temporal nature of repetitive consumption patterns, the model provides a more sophisticated way to account for both long-term and short-term user interests. The authors combine these approaches with a multi-level attention hybrid network to refine short-term interests and use a generalized EM framework to train the model. Results show that the approach improves the modeling of user intent, providing more personalized and accurate recommendations.

Quality:
This paper presents a high-quality model that integrates several state-of-the-art techniques, such as hypergraphs, Hawkes processes, and multi-level attention networks. The methodology is sound and comprehensive, and the proposed model appears to address the key challenges of modeling repetitive user intent over time.

Clarity:
The clarity of the paper is generally good, but could be improved in several areas. The introduction and motivation are well explained, but some of the more technical aspects—such as the use of Hawkes processes, hypergraphs, and the generalized EM framework—may need further explanation or clarification to ensure that a wider audience can understand them.

Originality:
The originality of this paper lies in the integration of several sophisticated techniques to model user intent in sequential recommender systems. The use of hypergraphs and Hawkes processes to capture both short-term and long-term user interests, especially focusing on repetitive consumption patterns, is a novel approach. While sequential recommender systems usually focus on short-term or long-term preferences respectively, this paper provides an innovative approach to model both simultaneously, which can lead to more accurate and personalized recommendations.

Significance:
This work is of great significance, especially in the case where users exhibit repetitive consumption behavior over time. Many traditional recommender systems have difficulty accounting for such repetitive intent, especially when user preferences evolve or repeat periodically. The HyperHawkes model provides a sophisticated framework to address this challenge, making it highly relevant to applications such as e-commerce, streaming services, and other fields where user behavior is cyclical or repetitive.

pros:
Innovative approach: Combining hypergraphs, Hawkes processes, and multi-level attention networks to model user intent in sequential recommendations is a novel and impactful contribution.
Well-structured approach: The integration of multiple advanced techniques into a unified framework provides a powerful approach to handle user intent in recommender systems.

cons:
1. The method is technically complex, and some concepts (such as Hawkes process and hypergraph-based modeling) may be difficult for readers without background in these fields to understand.Hawkes process lacks a specific formula.The article claims to be a generalized EM algorithm, but lacks specific and rigorous proof
2. The article lacks comparison with newer models when selecting baseline models, such as [rf1] and [rf2]

[rf1]: Multi-Behavior Hypergraph-Enhanced Transformer for Sequential Recommendation
[rf2]: Session-based Recommendation with Hypergraph Attention Networks

**Questions:**

1. Can you provide more specific formulas and explanations about Hawkes process to help readers without relevant background understand this concept?
2. The method you mentioned is a generalized EM algorithm. Can you provide more detailed theoretical proof or derivation process to enhance the rigor and credibility of this method?
3. When selecting the baseline model, have you considered comparing with newer models? If not, could you explain the reasons for choosing the current baseline model and explain why no newer models were chosen for comparison?

**Reviewer Confidence:**

3: The reviewer is confident but not certain that the evaluation is correct

**Scope:**

4: The work is relevant to the Web and to the track, and is of broad interest to the community

---

### Official Review · Reviewer_JpBi · 2024-12-02

**Novelty:** 6
**Technical Quality:** 6

**Review:**

The paper introduces the HyperHawkes model, which leverages hypergraphs and temporal dynamics to effectively capture and recommend user intents in sequential recommendation systems, particularly focusing on repeated consumption patterns.

Strengths
1.Innovative Approach: The paper introduces the HyperHawkes model, which effectively combines hypergraph structures with the Hawkes process to model user intents and temporal dynamics.  This innovative approach enhances the understanding of user behavior, particularly in capturing repeated consumption patterns.
2.Comprehensive Evaluation: The authors conduct extensive experiments across multiple real-world datasets, demonstrating the model's superiority over various state-of-the-art methods.  The reported improvements in metrics such as HR@20 and NDCG@20 provide strong evidence of the model's effectiveness.
3.Ablation Study: The inclusion of a detailed ablation study allows for a clear understanding of the contributions of different components within the HyperHawkes model.  This transparency strengthens the validity of the findings and highlights the importance of both short-term and long-term interest modeling.
4.Practical Implications: The model's ability to effectively capture user intents and improve recommendation performance has significant implications for real-world applications in recommendation systems, making it relevant for both academia and industry.

**Questions:**

1.The authors fail to analyze the model's complexity from a theoretical perspective, considering only the training time per epoch on the dataset as a factor for model complexity.
2.The performance of the HyperHawkes model is heavily reliant on the availability of rich and high-quality user interaction data.  In cases where data is sparse or noisy, the model's effectiveness may be significantly diminished, limiting its applicability in certain domains.
3.The hyperparameters involved in the model need to be carefully tuned for different datasets, which may increase the complexity and time cost of applying the model.
4. Due to the complexity of the model, the training and inference process may require high computational resources, which can be a hindrance in resource-limited environments.

**Reviewer Confidence:**

3: The reviewer is confident but not certain that the evaluation is correct

**Scope:**

4: The work is relevant to the Web and to the track, and is of broad interest to the community

---

### Official Review · Reviewer_TNAD · 2024-12-02

**Novelty:** 5
**Technical Quality:** 5

**Review:**

This paper presents a sophisticated approach to enhancing sequential recommendation systems by modeling user intents through hypergraphs and temporal dynamics.The introduction of the HyperHawkes model, which integrates hypergraphs and Hawkes processes, demonstrates a comprehensive understanding of both the theoretical and practical aspects of recommendation systems. The methodology is well-defined, and the use of contrastive clustering for intent extraction adds depth to the analysis. Furthermore, the experimental results indicate that HyperHawkes significantly outperforms existing state-of-the-art methods across multiple datasets, showcasing its effectiveness.

Pros:
1. This paper introduces a novel HyperHawkes model that effectively combines hypergraphs and temporal dynamics for better intent modeling.
2. Demonstrates significant improvements over existing models in various real-world datasets, indicating practical applicability.

Cons:
1. As the number of users and items increases, the computational demands of the hypergraph construction and the Hawkes process modeling can become significant. This may lead to scalability issues, particularly in real-time recommendation systems where quick response times are essential.
2. User preferences can change rapidly, and the model may not adapt quickly enough to these shifts without additional mechanisms for dynamic updating.

**Questions:**

The same as above.

**Reviewer Confidence:**

3: The reviewer is confident but not certain that the evaluation is correct

**Scope:**

3: The work is somewhat relevant to the Web and to the track, and is of narrow interest to a sub-community

---

### Official Review · Reviewer_r6K2 · 2024-12-04

**Novelty:** 5
**Technical Quality:** 6

**Review:**

# Summary
This paper proposes a HyperHawkes model which use Hawkes Processes to model the temporal dynamics of intents and a multi-level attention mixture network to fuse long-term and short-term signals.

# Pros
1. The paper is easy to follow.
2. Comprehensive experiments and I do not see any base comparison missing. Numerous ablation study shows the efficiency of each components.

# Cons
1. The combination of long-term intents and short-term interests can be further discussed. For instance, the paper might consider representing each aspect with dense embeddings and fusing them using multi-layer perceptrons (MLPs).

**Questions:**

see the cons above

**Reviewer Confidence:**

4: The reviewer is certain that the evaluation is correct and very familiar with the relevant literature

**Scope:**

4: The work is relevant to the Web and to the track, and is of broad interest to the community